# Obstetric Violence among Pregnant Jordanian Women: An Observational Study between the Private and Public Hospitals in Jordan

**DOI:** 10.3390/healthcare11050654

**Published:** 2023-02-23

**Authors:** Omar A. Azzam, Amer Mahmoud Sindiani, Maysa M. Eyalsalman, Mira K. Odeh, Kenda Y. AbedAlkareem, Sara A. Albanna, Elaf M. Abdulrahman, Weaam Q. Abukhadrah, Haitham O. Hazaimeh, Ashraf Ahmed Zaghloul, Samir S. Mahgoub

**Affiliations:** 1Department of Obstetrics and Gynecology, Faculty of Medicine, Mutah University, Al-Karak 61710, Jordan; 2Department of Obstetrics and Gynecology, Faculty of Medicine, Jordan University of Science and Technology, Irbid 22110, Jordan; 3Faculty of Medicine, Mutah University, Al-Karak 61710, Jordan; 4Faculty of Medicine, Al-Balqaa Applied University, Al-Salt 19117, Jordan; 5Faculty of Medicine, Jordan University of Science and Technology, Irbid 22110, Jordan; 6Department of Public Health, Faculty of Medicine, Mutah University, Al-Karak 61710, Jordan; 7Department of Health Administration, High Institute of Public Health, Alexandria University, Alexandria 5424041, Egypt; 8Department of Biochemistry and Molecular Biology, Faculty of Medicine, Mutah University, Al-Karak 61710, Jordan; 9Department of Biochemistry and Molecular Biology, Faculty of Medicine, Al-Minia University, Al-Minia 2431436, Egypt

**Keywords:** obstetric violence, maternal care, violation, disrespect and abuse, childbirth

## Abstract

Background: Obstetric Violence (OV) is a public health matter that affects women and their children with an incidence rate between 18.3–75.1% globally. The delivery institution of public and private sectors represents a potential factor contributing to OV. This study aimed to assess OV existence among sample of pregnant Jordanian women and its risk factors domains between public and private hospitals. Methodology: This is a case-control study including 259 recently delivered mothers from Al-Karak Public and Educational Hospital and The Islamic Private Hospital. A designated questionnaire including demographic variables and OV domains was used for data collection. Results: A significant difference was seen between patients delivering in the public sector compared to patients delivering the private sector in education level, occupation, monthly income, delivery supervision and overall satisfaction. Patients delivering in the private sector showed a significantly less physical abuse by the medical staff compared to patients delivering in the public sector, and patients delivering in a private room also showed a significantly less OV and risk of physical abuse compared to patients delivering in shared room. In public settings, medications information was lesser versus the private ones, additionally, there is significant association between performing episiotomy, physical abuse by staff and the delivery in shared rooms in private settings. Conclusion: This study showed that OV was less susceptible during childbirth in private settings compared to public settings. Educational status, low monthly income, occupation are risk factors for OV; also, features of disrespect and abuse like obtaining consent for episiotomy performance, delivery provision updates, care perception based on payment ability and medication information were reported.

## 1. Introduction

Obstetric violence (OV) was first defined in 2007 as: “the act of healthcare professionals to control both the reproductive practices and the women^’^s bodies, leading to the loss of patients’ autonomy and the inability to give free decision for their own bodies and sexual activities, which could give a negative impact after delivery and ongoing on the quality of women’ s life” [1]. The incidence rate of OV has been reported to be between 18.3–75.1% all over the world [2,3,4,5].

The World Health Organization (WHO) categorized OV into routine, unnecessary procedures and medications (on the mother and/or the infant), verbal abuse, humiliating act or physical violence, inadequate materials and facilities or complete absence of both, passing over the women’s permission along with providing insufficient information about the procedures to be performed by the professionals and the residents, and discrimination based on religious, economic, ethnic and cultural issues [6].

Obstetrical and gynecological situations like examinations, caesarean section and episiotomy may be considered as human rights violation, furthermore, could be experienced as forms of sexual acts and gender discrimination. The WHO has employed Disrespect and Abuse organization (D&A) [7] for elucidating this problem by focusing on birthing women rights, it protects freedom and equality and defends against any form of discrimination, sexual violence, verbal, mental and physical abuse, its protocol includes an agreement between staff and mothers to receive the highest level of care in a programmed time [8].

Disrespect and abuse could be identified in seven categories (non-consented care, non-confidential care, discrimination, non-dignified care, detention in facilities physical abuse and abandonment of care) which may maximize the susceptibility of women to an intentionally or non-intentionally mistreatment [9]. Other mistreatment situations were studied, like carrying out the procedure of non-anesthetized episiotomy would be put in consideration as an abusive act in all countries and lead to understand the meaning of mistreatment during childbirth, which is variable in different cultures based on respect, disrespect, and abuse concept [8].

Different forms of mistreatment and abuse during childbirth have been reported to be a well-known phenomenon in several parts of the world including Asia, Africa and Latin America [10,11,12], most of the studies done in this field employed qualitative methodologies to learn about women’s experience regarding their mistreatment and abuse suffering [13].

Furthermore, most of the childbearing women have reported that they were not informed about the induction of labor, not being asked to give informed consent before the procedure and lacked the adequate support during delivery [14]. Women’s feelings of susceptibility during childbirth can be worsened by unsuitable behavior of the staff, poor infrastructure, the absence of healthcare resources and policies as a result of professional unawareness or in-service training [15].

Studies show that negativity of childbirth experience was associated with complications of childbirth such as post-partum hemorrhage, labor induction, maternal infection, injury of anal sphincter the fear of childbirth and the abuse history [16,17].

In 2019, a report issued by UN General Assembly suggesting a human rights approach directed against several mistreatment forms that women have underwent during childbirth by asserting on violating the rights of women to be in a non-violent environment, and imperiling their autonomy, health, rights to life and all rights in contradiction of discrimination [18,19].

OV can also be affected by delivery institutional factors, such as when the healthcare administration does not guarantee safe attention during labor [20]. Furthermore, it is under effects of legal and political issues when women are not given long maternity leave for supporting their babies emotionally, physically and to breastfed them during the first six month of life [21]. However, no studies to our knowledge have yet investigated the effect of delivery institutions on the prevalence of OV.

The aim of the study was to assess the prevalence, and risk factors of OV domains among pregnant Jordanian women between public and private hospitals in Jordan and assess the difference of OV in both sectors. To the best of our knowledge, this is the first study to be conducted in Jordan on OV.

## 2. Martials and Methods

### 2.1. Study Setting

The study was conducted at one public and one private hospitals in Jordan: Al-Karak educational and public hospital, a 430 bed-hospital in Al-Karak, and The Islamic private hospital, a 400 bed-hospital in Amman.

### 2.2. Study Design

A case-control study design was followed to answer the study question. Based on the hypothesis of exposure to obstetric violence is expected to occur in public hospitals. Cases were termed as mothers delivered in public hospitals; controls were termed as mothers delivered at private hospitals. Patients did not receive any remuneration, and all collected data were anonymized.

### 2.3. Study Population

The study included mothers who have recently delivered at the above-mentioned hospitals in Jordan.

### 2.4. Sampling Technique

The study was conducted from the beginning of December 2021 till the end of February 2022. A convenient sample of delivered mothers from both public and private hospitals of Amman was included in the study and agreed to participate in the study.

### 2.5. Data Collection Tool

After reviewing the literature on obstetric violence, a questionnaire including the domains of obstetric violence was designed. The tool included two sections: section the demographic characteristics of the sample under study. The demographic variables included age (quantitative continuous variable) which was eventually categorized into two groups (<30 years and 30+ years). The level of education variable (categorical variable consisted of two groups namely; read and write/school education, and university education, occupation (categorical variable consisted of two groups namely; house/unemployed and employed group). The variable monthly income (categorical variable consisted of two categories; less than 500 JOD and 500+ JOD. The variable delivery supervision (categorical variable consisted of two groups namely; physician and nurse/midwife groups). The variable of overall satisfaction with the experience of delivery at the hospital (categorical variable consisted of two groups namely, satisfied and dissatisfied groups). The second section of the tool consisted of questions related to the domains of obstetric violence which included: feeling of respect, episiotomy, episiotomy consent, medication information, physical abuse by medical staff, delivery updates, feeling of discrimination, perception of care based on ability to pay. The responses to the above-mentioned questions were in the form of binary options (Yes = 0, No = 1). The response for the question type of delivery room was in the form of options (shared = 0, private = 1).

### 2.6. Data Analysis

Data were collected, organized, coded, and checked for missing or irrelevant data items. Results were presented in tabular form of frequencies and percentages. All variables were described as frequencies (Percentage %). Correlations and associations between categorical variables were tested using the Chi-squared χ^2^ test or Fisher exact test if a category count was <5. A binary logistic regression model was used to test the risk factors associated with OV domains in patients who underwent episiotomy. The domains were checked for multicollinearity using the variance inflation factor. Tests of significance were adjusted at the 5% level of significance to defy the null hypothesis of the study.

## 3. Results

The study sample mean age in both settings was 31.2 ± SD 8 years (Table 1). The age group 30+ years was highly represented in both groups, public setting (62.3%) and private setting (50.5%). The read and write/school education was highly represented at the public setting (67.3%) and university education at private settings (76.3%), a statistically significant association was detected between education and settings (χ^2^ = 46.09). There was a significant difference in education levels between patients delivering at public vs. private hospitals, in which 76.3% of patients delivering at private sectors had a university education while only 32.7% of patients delivering at public sectors had a university education. 

In public and private settings, the highly represented group was the same for occupation of housewife/unemployed (88.9% and 63.9%, respectively), delivery supervision for physician (78.4% and 90.7%, respectively), and overall satisfaction for satisfied (65.4% and 77.3%, respectively). A statistically significant association was detected for the demographic variables of the sample under study (occupation χ^2^ = 23.24, delivery supervision χ^2^ = 6.53, overall satisfaction χ^2^ = 4.07, respectively). The highest percentage of monthly income less than 500 JOD were represented in the public settings (80.9%) whereas the highest percentage of monthly income equal or greater than 500 JOD were represented at private settings (53.6%). A statistically significant association was detected (χ^2^ = 33.1).

Table 2 represents the distribution of the obstetric violence domains at both public and private setting of the sample under study. The highest percentage of groups at both public and private settings was represented for feeling of respect (74.1% and 85.6%, respectively), no consent for the performance of episiotomy (62.3% and 51.3%, respectively), no physical abuse by the medical staff (78.4% and 92.8%, respectively), provision of delivery updates (60.5% and 70.1%, respectively), feeling of discrimination (76.5% and 86.6%, respectively) and delivery in a private room (54.9% and 87.6%, respectively). The highest percentage of not securing an episiotomy consent was represented at both public and private settings, respectively (73.7% and 79.3%, respectively). Whereas the highest percentage of not providing medication information was represented at public settings (51.2%) while the highest percentage for providing medication information was represented at private settings (58.8%). A statistically significant correlation was detected between no physical abuse by medical staff and delivery in the private sector, in addition to delivery in a private room compared to shared room (χ^2^ = 9.24 and 29.41, respectively).

Table 3 represents the odds ratio for the domains of obstetric violence in episiotomy cases in both public and private settings. Women who performed episiotomy at public settings were significantly less like to feel respect (OR = 0.7; 95% CI: [0.5–0.9]). No significant odds were detected for medication information (OR = 0.8; 95% CI: [0.6–1.2]), less likely to receive delivery updates (OR = 0.7; 95% CI: [0.6–1.2]), securing an episiotomy consent (public settings OR = 1.1 and private settings OR = 0.8), feeling of discrimination (OR = 1.2, CI = 0.8, 1.7) and perceive that care was based on the ability to pay (OR = 1.3; 95% CI: [0.9–1.8]). Women who performed episiotomy at public settings were significantly more likely to perceive physical abuse (OR = 1.6; 95% CI: [1.2–2.1]) and were more likely to delivery in shared rooms (OR = 1.8; 95% CI: [1.4–2.5]).

## 4. Discussion

Obstetric violence during childbirth can be understood based on the concept of disrespect to women’s autonomy, feelings, mental integrity, physical reliability, and abuse for identifying the directed violence act against pregnant women or her baby [22]. It can be considered as a phenomenon documented through various violence situations during gestation, delivery, puerperium, as well as the assisted cases like the reproductive cycle, miscarriage and post-miscarriage [23,24].

The term disrespect and abuse was suggested to identify any violence act during the professional assistance of childbirth, it describes care aspects comprising of non-dignified care, non-confidential care, non-consented care, facilities detention, discrimination built on precise patients’ attributes, physical abuse, and care abandonment [25,26].

The WHO assured that low socioeconomic, ethnic minorities, adolescents, single women, are mostly subjected to suffer from disrespect and abuse [4]. The knowledge, the understanding of the concept “disrespect and abuse during childbirth”, and the respect experience are crucial to design strategies for strengthening the systems providing respectful care [27]. On the other hand, the alertness women of their rights and who have never been exposed to any further care system are not sensitive to the health care employees^,^ disrespect, and abuse [5].

In the present study, the results revealed that women of the read/write school and the university educational levels were higher in the public and private settings, respectively, a significant association was detected between the educational level and the settings. Globally, the incidence of OV was reported to be 15–97% globally with greater OV risk to the deprived women with lower educational level even in industrialized countries [28,29], hence, OV had been predicted and associated significantly to various attributable factors including educational status.

In previous studies, OV was demonstrated to be predicted by the educational level [30,31], which is significantly associated with OV, it was reported that women who attended low educational level are lesser than who attended higher educational level regarding their capability to report for OV, in fact the highly educated women are more alert of their rights and have the tendency to report any OV form they may be subjected to [32]; also, the monthly income of the family was considered [32].

Regarding delivery supervision for physician and overall satisfaction for satisfied were higher in the private setting, while it was lesser for occupation of housewife/unemployed when compared to the public setting. The obtained results showed significant association for occupation, delivery supervision and satisfaction. Furthermore, a statistically significant association was found in the monthly income which was equal or higher in the private setting than the public setting.

Concerning the different domains of OV, our results showed that the feeling of respect, not being subjected to any physical abuse by residents and professionals, delivery updates provision, feeling of discrimination and the delivery in a private room were higher in the private settings versus the public settings, while it was lesser regarding informing the pregnant woman during delivery about the performance of episiotomy. Delivery in a private room and the absence of physical abuse by medical staff were significantly associated.

During childbirth, women are in need for privacy without unnecessary procedures because of being not favored, they wanted intervention with no cuts. Exposure, vaginal examinations, episiotomy, and its repair can be considered abusive and disrespectful shameful act as stated by women [5]. In another study, the performance of episiotomy without taking an informed consent was identified as an aggressive practice without being considered as an OV [27].

The results obtained in the present study showed that women subjected to the performance of episiotomy during childbirth at public settings were significantly less likely to feel respect, in obtaining an informed episiotomy consent, to be informed about the delivery updates and the medications they receive (no significant odds), feeling of discrimination and the ability to pay is an indicator for perceiving the required maternal care, while, in public settings, significant association was noticed between the performance of episiotomy, perceiving physical abuse and the delivery in shared rooms (OR = 1.6, 1.8, respectively). Those findings are in line with the WHO study in four African countries which detect that about 30% of the women reported about experiencing disrespect and abuse during delivery their babies and the young women were more likely to experience physical abuse, the association between the low educational level and verbal abuse was prominent [33].

In another study conducted in North Showa Zone, Ethiopia, more than 40% of the participants seeing different forms of physical abuse such as slapping, pinching or beating due to the noise they have made or being uncooperative; also, they reported the performance and suturing of episiotomy without anesthesia, even not allowance of the favored birthing position and beating with the instruments were reported [34]. Furthermore, less or even no existing verbal and physical abuse were noticed in the private setup versus the public setup which was attributed to the available facilities and the small number of patients, this is in agreement with what we have obtained in the present study [35]. In the study of Anna and Hafrún [36], found that the participated women had experienced both psychological and physical abuse including threats of violence and the birth experience was expressed as compared to rape.

Limitations to the study include the small sample size and the inability to add more questions regarding the religion, ethnicity and the nationality of the participants. Furthermore, the study was based on interviews not the contact observation.

## 5. Conclusions

Women in the present study experienced respect, disrespect with different degrees in the public and the private settings, though the exploration of the perceptions of OV domains. There were some aspects of mistreatment or disrespectful act from the health services professionals directed against women during childbirth. A significant association was shown between the educational situation and the settings in the study; also, for the demographic variables (occupation, the overall satisfaction, delivery under healthcare workers’ supervision and for the monthly income of the family. The highest percentages of the presented OV domains were for the private settings except for taking the consent for the episiotomy performance, it was higher for the public settings, a statistically significant association was detected between the no physical abuse by the healthcare professionals and giving birth in private room not the shared one. In the public settings, it was shown that women subjected to the episiotomy procedure during childbirth were significantly less likely to have the impression of respect, in being requested for episiotomy written or verbal informed consent, updating them for the delivery and the medications they are given, touching any form of discrimination and the capability for the payment to receive the optimal maternity care, an association was found between the episiotomy performed the presence of physical abuse and the childbirth in shared rooms.

## Figures and Tables

**Table 1 healthcare-11-00654-t001:** Demographic variables at public and private settings of the sample under study (Amman, 2022).

Demographic Variables	Public Setting (*n* = 162)		Private Setting (*n* = 97)		χ^2^
	No.	%	No.	%	
**Age group**					
Less than 30 years	61	37.7	48	49.5	3.48
30+ years	101	62.3	49	50.5	
**Education**					
R&W/School education	109	67.3	23	23.7	46.09 *
University education	53	32.7	74	76.3	
**Occupation**					
Housewife/Unemployed	144	88.9	62	63.9	23.2 *
Employed	18	11.1	35	36.1	
**Monthly income**					
Less than 500 JOD	131	80.9	45	46.4	33.1 *
500+ JOD	31	19.1	52	53.6	
**Delivery supervision**					
Physician	127	78.4	88	90.7	6.53 *
Nurse/Midwife	35	21.6	9	9.3	
**Overall experience satisfaction**					
Satisfied	106	65.4	75	77.3	4.07 *
Dissatisfied	56	34.6	22	22.7	

** p* < 0.05.

**Table 2 healthcare-11-00654-t002:** Obstetric violence in public and private settings (Amman, 2022).

Obstetric Violence	Public Setting (*n* = 162)		Private Setting (*n* = 97)		χ^2^
	No.	%	No.	%	
**Feeling of respect**					
Yes	120	74.1	83	85.6	4.72
No	42	25.9	14	14.4	
**Episiotomy**					
Yes	61	37.7	47	48.5	2.91
No	101	62.3	50	51.5	
**Episiotomy consent**					
Yes	25	26.3	18	20.7	0.79
No	70	73.7	69	79.3	
	(*n* = 95)		(*n* = 87)		
**Medication information**					
Yes	79	48.8	57	58.8	2.43
No	83	51.2	40	41.2	
**Physical abuse by medical staff**					
Yes	35	21.6	7	7.2	9.24 *
No	127	78.4	90	92.8	
**Delivery updates**					
Yes	98	60.5	68	70.1	2.43
No	64	39.5	29	29.9	
**Type of delivery room**					
Shared	73	45.1	12	12.4	29.41 *
Private	89	54.9	85	87.6	
**Feeling of discrimination**					
Yes	38	23.5	13	13.4	3.87
No	124	76.5	84	86.6	
**Perception of care based on ability to pay**					
Yes	91	56.2	56	57.7	0.06
No	71	43.8	41	42.3	

** p* < 0.05.

**Table 3 healthcare-11-00654-t003:** Risk of obstetric violence domains in episiotomy cases (*n* = 108) in public to private settings (Amman, 2022).

Obstetric Violence	Public Setting (*n* = 61)			Private Setting (*n* = 47)		
	OR	95% CI		OR	95% CI	
		LL	UL		LL	UL
**Feeling of respect**						
Yes	0.7	0.5	0.9	1.7	0.9	3.1
No						
**Episiotomy consent**						
Yes	1.1	0.7	1.5	0.8	0.5	1.4
No						
**Medication information**						
Yes	0.8	0.6	1.2	1.1	0.76	1.8
No						
**Physical abuse by medical staff**						
Yes	1.6	1.2	2.1	0.4	0.2	0.9
No						
**Delivery updates**						
Yes	0.7	0.6	1.2	1.3	0.8	2.3
No						
**Type of delivery room**						
Shared	1.8	1.4	2.5	0.3	0.1	0.6
Private						
**Feeling of discrimination**						
Yes	1.2	0.8	1.7	0.7	0.3	1.3
No						
**Perception of care based on ability to pay**						
Yes	1.3	0.9	1.8	0.7	0.4	1.1
No						

## Data Availability

All data associated with this study are present in the paper.

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
