# Peer review of "Obstetric Violence among Pregnant Jordanian Women: An Observational Study between the Private and Public Hospitals in Jordan"

_healthcare, 2023, doi:10.3390/healthcare11050654_

Round 1

Reviewer 1 Report

Please see attached review. 

Author Response

Reviewer 1:

  1. Line 29-30 - confusing statement; what does lack of physical abuse entail in this context?

- Response: The lack of physical abuse in the context of the abstract entailed those patients delivering in the private sector showed a significantly less physical abuse by medical staff than patients delivering in public sectors, in addition to patients delivering in a private room showed significantly less OV and physical abuse compared to patients delivering in a shared room.

As the sentence in the abstract seems ambiguous, we have changed it in both the abstract and the results section as follows:

Abstract:

Patients delivering in the private sector showed a significantly less physical abuse by the medical staff compared to patients delivering in the public sector, and patients delivering in a private room also showed a significantly less OV and risk of physical abuse compared to patients delivering in shared room.

Results:

A statistically significant correlation was detected between no physical abuse by medical staff and delivery in the private sector, in addition to delivery in a private room compared to shared room (X2= 9.24 and 29.41 respectively).”

  1. The introduction's definition of OV is too general and redundant. In this instance, the majority of readers may not have grasped what OV is. I respectfully advise that the writers utilize an integrative method rather than a SILO approach so that readers can understand it easily. In general, the introduction should be reviewed again with special attention to how each paragraph flows into the next.

- Response: Thank you for pointing this out. As suggested by the reviewer, we have made changes throughout the introduction sections to make it clearer and reader-friendly. All redundant information has been removed and paragraphs have been ordered to meet a better flow.   

  1. When analyzing data, I advise authors to be clear about the analysis (for example, chi square, regression?).

- Response: Thank you for illustrating this point. All the statistical test used have been added and cleared accordingly in the data analysis section of the methods.

The revised text reads as follows:

“2.6. Data analysis

Data were collected, organized, coded, and checked for missing or irrelevant data items. Results were presented in tabular form of frequencies and percentages. All variables were described as frequencies (Percentage %). Correlations and associations between categorical variables were tested using Chi-squared χ2 test or Fisher exact test if a category count was <5. A binary logistic regression model was used to test the risk factors associated with OV domains in patients who underwent episiotomy. The domains were checked for multicollinearity using the variance inflation factor. Tests of significance were adjusted at the 5% level of significance to defy the null hypothesis of the study.”

  1. Lines 100 and 101 --- the purpose of the study was to determine whether pregnant Jordanian women receiving care in both public and private institutions had OV domains and what their risk factors were. - However, Table 1 compares demographics for the public and private sectors. Which part of the study's goal is Table 1 trying to address?

- Response: The aim of the study was vaguely described in the introduction section. The amended part of the introduction clarifies that the objective is to study the prevalence of OV between the private and public sectors and the demographic variables and OV domains between patients delivering in public sectors vs. patients delivering in private sectors and identify risk factors associated with OV. Thus, comparison between patients delivering in public sectors and patients delivering in private sectors is necessary to meet the objective of the study.

The revised text regarding the objective of the study in the introduction section presents as follows:

“The aim of the study was to assess the prevalence, and risk factors of OV domains among pregnant Jordanian women between public and private hospitals in Jordan and assess the difference of OV in both sectors. To the best of our knowledge, this is the first study to be conducted in Jordan on OV.”

  1. In Table 2, under the heading "Obstetric violence at public and private settings (Amman, 2022)," writers compare public and private in the context of OV ---It is unclear in what perspective to look at the relationship between the private and public in relation to OV domains. Is it not required that OV (in terms of domain) exist in both the private and public spheres? Do the writers compare the presence of OV to determine if it is more substantial (private or public) in one place vs the other?

- Response: As we illustrated in the previous point, the study first aimed to investigate the difference if OV domains between in both public and private sectors, and possible risk factors associated with the OV depending on the delivery sector. As we first studied the difference in demographic characteristics of patients delivering in public vs. private sectors (Table 1), then study the difference in prevalence and domain between the two sectors (Table 2 & 3).

  1. Do the authors indicate that the domains affect the risk of OV? Are the demographics not meant to be the concern? For instance, may the monthly income of expectant mothers in both private and public hospitals pose a risk to OV?

- Response: Thank you for illustrating this point. Yes, OV domains show to affect the risk of OV, as seen in Table 3, those domains reflect the initiation or development of OV. As patients who did not observe OV, tend to have better results regarding the OV domains. However, demographic characteristics were studied as a driver for patients choosing which sector. As shown in Table 1, patients with university education level, tend to choose private sectors for delivery, which in turn reflects the prevalence of OV in this sector.

Reviewer 2 Report

                             Reviewer's  Comments

                                 01-16-2023

1-The title “Obstetric Violence among Pregnant Jordanian Women” does not completely capture the main content of the present study. The authors should add a subtitle"-Take XXX hospital as An Example".

2-The authors do not know how to write an abstract. The abstract should begin with a brief but precise statement of the problem or issue, followed by a description of the research method and design, the major findings, and the conclusions reached. For example, the abstract could be written in the following manner: With convenience sampling approach, 259 pregnant women in XX hospital and XXX hospital in Jordanian were selected with convenience sampling method to investigate the status quo of XXX,XXX,XXX, and the impacting factors with descriptive statistics and inferential statistics. The results showed that blah, blah, and blah…. (Add statistical analysis results to the corresponding places).

3-The depth of the literature review in the present study is not enough. So, the reviewer failed to see the research gap between the proposed study and the existing studies in this field. As the authors of the study, you should ask yourself several questions below—What are new in your key findings? What this adds to what is known? What is the implication, what should change now? please rewrite the section “Introduction”.

4-From line 100 to line 102, the authors described the research purpose. However, as a scientific research paper, where is or are research question(s)? Imagine how could readers follow the writing to keep reading the manuscript and understand the issues that the authors try to investigate. Please explicitly formulate research questions of the study. For example, is there a difference between groups? 

5-It seems that there are no missing values on the background variables of the study. However, there are no relevant description of other variables. The authors should clarify if it is the case. What is the rate and pattern of missing values? Is the pattern of missing values MCAR or MNAR?

6-Did the participants received the remuneration as an incentive?

7-What are the inclusion and exclusion criteria for the participants in the present study?

8-Does the collection reflect methods that produce unbiased results? More specifically, the authors should provide the information about who collected the data. Also, the authors should discuss quality assurance procedures in data collection and verification and how data collection settings or methods have influenced the data collected. Did the present study use online survey administration System like “Qualtrics” or traditional pencil-paper questionnaire? These questions need to be clarified.

9-% and n in Table 1 and Table 2 should be replaced with percent and frequency, respectively.

10-X2 is incorrect representation of chi-square value. The correct one should use Greek Letter.

11-How did the authors justify the sample size of this study?

12-What is the effect size of chi-square test? How do the authors interpret it?

13- Did the authors check the assumption of binary logit regression model?

14-Did the authors check the over-dispersion?

15-How did the authors justify the linear type of the logit model? Why not quadratic or triple terms?

16-Re-write the discussion and conclusion part after correcting the problems above.

Author Response

Reviewer 2:

  1. The title “Obstetric Violence among Pregnant Jordanian Women” does not completely capture the main content of the present study. The authors should add a subtitle"-Take XXX hospital as An Example

- Response: Thank you for this suggestion. The title has been modified to “Obstetric Violence among Pregnant Jordanian Women: An observational study between the private and public hospitals in Jordan.”

  1. The authors do not know how to write an abstract. The abstract should begin with a brief but precise statement of the problem or issue, followed by a description of the research method and design, the major findings, and the conclusions reached. For example, the abstract could be written in the following manner: With convenience sampling approach, 259 pregnant women in XX hospital and XXX hospital in Jordanian were selected with convenience sampling method to investigate the status quo of XXX,XXX,XXX, and the impacting factors with descriptive statistics and inferential statistics. The results showed that blah, blah, and blah. (Add statistical analysis results to the corresponding places).

- Response: Thank you for this suggestion. The revised abstract has been amended to reflect the study flow and provide a better representation of the results.

  1. The depth of the literature review in the present study is not enough. So, the reviewer failed to see the research gap between the proposed study and the existing studies in this field. As the authors of the study, you should ask yourself several questions below—What are new in your key findings? What this adds to what is known? What is the implication, what should change now? please rewrite the section “Introduction”.

- Response: Thank you for pointing this out. The study addressed the gap of OV between public and private sectors and the associated risk factors which is to be the first study in Jordan. As suggested by the reviewer, the introduction section has been modified to be more presentative of the research objective and address the research gap. 

  1. From line 100 to line 102, the authors described the research purpose. However, as a scientific research paper, where is or are research question(s)? Imagine how could readers follow the writing to keep reading the manuscript and understand the issues that the authors try to investigate. Please explicitly formulate research questions of the study. For example, is there a difference between groups?

- Response: Thank you for pointing this out. The purpose of the study was vaguely describe not representing the research question. We have reformulated the question to be understandable in the last paragraph of the introduction section.

The revised part presents as follows:

“The aim of the study was to assess the prevalence, and risk factors of OV domains among pregnant Jordanian women between public and private hospitals in Jordan and assess the difference of OV in both sectors. To the best of our knowledge, this is the first study to be conducted in Jordan on OV.”

  1. It seems that there are no missing values on the background variables of the study. However, there are no relevant description of other variables. The authors should clarify if it is the case. What is the rate and pattern of missing values? Is the pattern of missing values MCAR or MNAR?

- Response: There were no missing values in any variable. But only subset of patients (182/259) from both sectors underwent episiotomy and were analyzed separately for the risk of OV as in Table 3.

  1. Did the participants received the remuneration as an incentive?

- Response: Patients did not receive any remuneration and consent was obtained following ethical approvals and institutional review boards (IRB) regulations. We have added this statement to the study design section in the methods to be clearer as suggested.

The amended part reads as follows:

“2.2. Study design

A case-control study design was followed to answer the study question. Based on the hypothesis of exposure to obstetric violence is expected to occur in public hospitals. Cases were termed as mothers delivered in public hospitals; controls were termed as mothers delivered at private hospitals. Patients did not receive any remuneration and all collected data were anonymized.”

  1. What are the inclusion and exclusion criteria for the participants in the present study?

- Response: As stated in the study population section of the methods, enrollment was for all mothers who delivered at Al-Karak Educational and Public hospital and Islamic Private Hospital between the period of December 2021 till the end of February 2022. There was no other inclusion or exclusion criteria to be set that would alter the objective of the study.

  1. Does the collection reflect methods that produce unbiased results? More specifically, the authors should provide the information about who collected the data. Also, the authors should discuss quality assurance procedures in data collection and verification and how data collection settings or methods have influenced the data collected. Did the present study use online survey administration System like “Qualtrics" or traditional pencil-paper questionnaire? These questions need to be clarified.

- Response: Thank you for illustrating this point. An online questionnaire was designed based on the literature for the aforementioned domains and was implemented as the data collection method. Response bias was avoided by designing the questionnaire to meet high quality by specifying key answers and allowing unbiased answers of 1 and 0. In addition, questionnaires provide some advantages over interviews in that they allow for anonymity which increases the likelihood of truthful and accurate responses. They also ensure consistency that all participants are asked the same questions, thus increasing the reliability of the data collected. Data collectors were stated in the author’s contribution and acknowledgement sections.

  1. % and n in Table 1 and Table 2 should be replaced with percent and frequency, respectively.

- Response: N and % frequency have been replaced with frequency and percentage respectively in the tables as suggested by the reviewer.

  1. X2 is incorrect representation of chi-square value. The correct one should use Greek Letter.

- Response: As suggested by the reviewer, the symbol of Chi-squared test has been corrected to the Greek letter χ2 where applicable in the manuscript.

  1. How did the authors justify the sample size of this study?

- Response: The sample size of the study was adequate and representative as the data were collected from two hospitals in Jordan and different sectors (both private and public) with a total of 259 participants.

  1. What is the effect size of chi-square test? How do the authors interpret it?

- Response: The effect size value  for the chi-squared test was not used in reporting the statistical significance of the associations, rather the chi-squared value χ2 was used to represent the significance of the association. However, the odds ratio (OR) from the logistic regression model was used to represent the effect size of the association as in Table 3.

  1. Did the authors check the assumption of binary logit regression model?

- Response: Thank you for illustrating this point. The assumptions for the binary logistic regression model have been met and tested for multicollinearity using the variance inflation factor. As suggested by the reviewer, the binary logistic regression analysis criteria have been added to the data analysis section of the methods.

The revised text reads as follows:

“2.6. Data analysis

Data were collected, organized, coded, and checked for missing or irrelevant data items. Results were presented in tabular form of frequencies and percentages. All variables were described as frequencies (Percentage %). Correlations and associations between categorical variables were tested using Chi-squared χ2 test or Fisher exact test if a category count was <5. A binary logistic regression model was used to test the risk factors associated with OV domains in patients who underwent episiotomy. The domains were checked for multicollinearity using the variance inflation factor. Tests of significance were adjusted at the 5% level of significance to defy the null hypothesis of the study.”

  1. Did the authors check the over-dispersion?

- Response: Over-dispersion was checked as an assumption for the binary logistic regression model based on the variance.

  1. How did the authors justify the linear type of the logit model? Why not quadratic or triple terms?

- Response: The linear form of logistic regression was chosen based on the relationship between the independent and dependent variables and the residual plot. The data analysis section of the methods has been revised to add information regarding the logistic model.

  1. Re-write the discussion and conclusion part after correcting the problems above

- Response: Since no major changes have been made to the results of the study, the discussion and conclusion did not need to be changed.

Round 2

Reviewer 1 Report

Dear authors, thank you for patiently revising the paper. Although some of the sentences can still be improved and some were mispelled(e.g. Materials and methods) but this can be acceptable. 

Reviewer 2 Report

Congratulation! This manuscript just needs to correct some minor English errors and then can be published.